# Developing patient-centred, feasible alternative care for adult emergency department users with epilepsy: protocol for the mixed-methods observational 'Collaborate' project

Adam J Noble ⬡,[1] Amy Mathieson,[1] Leone Ridsdale,[2] EA Holmes,[3] Myfanwy Morgan,[4] Alison McKinlay ⬡,[5] Jon Mark Dickson ⬡,[6] Mike Jackson,[7] Dyfrig A Hughes,[3,8] Steve Goodacre,[9] Anthony G Marson[10]

For numbered affiliations see end of article.

**Correspondence to**
Dr Adam J Noble;
adam.noble@liverpool.ac.uk

## ABSTRACT

**Introduction** Emergency department (ED) visits for epilepsy are common, costly, often clinically unnecessary and typically lead to little benefit for epilepsy management. An 'Alternative Care Pathway' (ACP) for epilepsy, which diverts people with epilepsy (PWE) away from ED when '999' is called and leads to care elsewhere, might generate savings and facilitate improved ambulatory care. It is unknown though what features it should incorporate to make it acceptable to persons from this particularly vulnerable target population. It also needs to be National Health Service (NHS) feasible. This project seeks to identify the optimal ACP configuration.

**Methods and analysis** Mixed-methods project comprising three-linked stages. In Stage 1, NHS bodies will be surveyed on ACPs they are considering and semi-structured interviews with PWE and their carers will explore attributes of care important to them and their concerns and expectations regarding ACPs. In Stage 2, Discrete Choice Experiments (DCE) will be completed with PWE and carers to identify the relative importance placed on different care attributes under common seizure scenarios and the trade-offs people are willing to make. The uptake of different ACP configurations will be estimated. In Stage 3, two Knowledge Exchange workshops using a nominal group technique will be run. NHS managers, health professionals, commissioners and patient and carer representatives will discuss DCE results and form a consensus on which ACP configuration best meets users' needs and is NHS feasible.

**Ethics and dissemination** Ethical approval: NRES Committee (19/WM/0012) and King's College London ethics Committee (LRS-18/19-10353). Primary output will be identification of optimal ACP configuration which should be prioritised for implementation and evaluation. A pro-active dissemination strategy will make those considering developing or supporting an epilepsy ACP aware of the project and opportunities to take part in it. It will also ensure they are informed of its findings.

**Project registration number** Researchregistry4723.

### Strengths and limitations of this study

► Project seeks to identify the optimal configuration of an 'Alternative Care Pathway' for epilepsy which could minimise clinically unnecessary and/or avoidable unplanned health service use and facilitate improved outcomes.

► The novel co-production approach employed—which will involve stakeholders at each project stage and use Discrete Choice Experiments (DCE)—will allow service users to inform service design in a scientifically robust, non-tokenistic way and could provide a template for service design more generally.

► Whilst most of the work packages will involve stakeholders from across England, patient and carers for the DCE are being recruited from one region. This might limit generalisability.

► The ACP configuration identified will be developed for use in the NHS and so may not be immediately applicable to other health systems. It will also require evaluation via a subsequent project.

► The project's success will depend, in part, on service users from what is a particularly vulnerable population agreeing to participate and representative samples being generated.

## INTRODUCTION

### Context and drive for health service innovations

The National Health Service (NHS), like other health systems, is being asked to make cost savings, while improving care experience, outcomes and reducing health inequalities.[1 2] Emergency department (ED) visits and admissions for the ambulatory care sensitive condition epilepsy provide an opportunity where service innovations could help achieve such aims. We describe a mixed-methods project to maximise the likelihood of such innovations being beneficial to service users and society.

## Epilepsy and unplanned service use

With a prevalence of ~1%,[3] epilepsy is the UK's second most common serious neurological disorder. Annually, ≤20% of people with epilepsy (PWE) visit hospital ED.[4–6] They report more anxiety, seizures and perceived epilepsy stigma than those in the wider epilepsy population and live in more deprived areas[7–12]; ~20% have an intellectual disability.[13 14]

The annual cost to the NHS of these visits in England alone is ~£70–90M.[15–17] Costs are high because half of the visits result in hospital admission.[6 18–21] An unusually high re-attendance rate also inflates cost[22 23];≤60% of PWE re-attend ED within 12 months.[7]

Most PWE visiting ED do not attend with a life threatening or emergency presentation, (eg, status epilepticus, first seizure). Rather, projects like our National Audits of Seizure Management in Hospitals (NASH)[13 14] show that most have known, rather than new epilepsy, and present with states not requiring ED's full facilities. Leading presentations include someone who has experienced: (i) an uncomplicated seizure in line with their usual presentation; (ii) a seizure in public and cannot be 'left at scene'; and (iii) a self-terminating seizure, different to their usual presentation.[20]

## Unmet needs of PWE visiting ED

While the acute episodes leading PWE to visit ED do not typically require emergency care, the visits can be expressions of the person having received suboptimal ambulatory care and unmet needs. NASH[19], for instance, found that most (~65%) PWE visiting ED are not known to specialist epilepsy services and many were via their usual care provider, seemingly receiving outdated care. For instance, despite focal epilepsy being the most common epilepsy type, most likely to be refractory and often not best treated with the medication sodium valproate, NASH found it to be the most prescribed medication among ED attendees.

Coping with life in the context of epilepsy also requires an individual to accept their diagnosis and learn and adopt a range of self-management behaviours to prevent seizures and manage consequences. PWE visiting ED though and their significant others appear to have less knowledge about epilepsy and its management, including seizure first aid.[7 24–26] They might therefore benefit from enhanced self-management support, like that provided by epilepsy nurses. The reason that knowledge might be low in ED attendees is because there remains no routine course that all PWE can go on to learn about epilepsy (as there are for some conditions). People who have lower education levels appear to fare worst from this situation with them having been found to have the least epilepsy knowledge.[27 28]

That some PWE in the UK are receiving suboptimal care is well known, with there being longstanding challenges in ensuring that the PWE most in need of specialist care receive it. While trial evidence indicates ~70% of PWE can become seizure free, some evidence indicates only ~50% of PWE in the UK currently are,[29] with PWE in socially deprived areas faring the worst.[30] Factors contributing to the challenge are that the UK has a comparatively small specialist workforce[31] and that there is no national, incentivised system to identify those within the epilepsy population that might need stepped-up care. To further complicate matters, General Practitioners (GPs), who care for most PWE, have expressed low confidence in managing the condition.

ED visits by PWE can be considered opportunities to intervene. Unfortunately, under current arrangements, going to ED does not typically lead to PWE receiving ambulatory care improvements. Bodies, such as The National Institute for Health and Care Excellence (NICE)[2], recommend that when seizures are not controlled, a patient should be referred to specialist services (within 4 weeks) since this may improve outcomes, including rendering some seizure free.[32 33] Most (80%) PWE visiting ED are though not seen by a specialist during their attendance, their usual care providers may not be informed of the attendance[13] and most (60%) are not referred to a specialist for follow-up.

PWE living in the most deprived areas, as well as the elderly, are among the least likely to be referred on from ED.[34] The low referral rate may be due to some clinicians holding an incorrect nihilistic view that intractability is inevitable if seizure control is not obtained within a few years of therapy onset. Assumptions about the willingness of certain patient groups to attend clinics may also be being made.[35 36]

## An 'Alternative Care Pathway'

NHS policies[1 2 37] and publicity given to NASH's findings[15 38] created momentum to reduce visits for seizures and enhance patient outcomes. NICE[39] found no evidence on how to do this. However, one idea gaining traction is for ambulance services to assume a greater role.[40] Most (~90%) people visiting ED for seizures have been transported there by an emergency ambulance.[13 14 21]

Data from some regional ambulance services on conveyance rates for seizures has been published. It indicates ambulance staff are recommending conveyance of nearly every person they attend for a suspected seizure to ED,[41–43] despite most not demonstrating a clinical need (eg, seizures have self-terminated before ambulance arrival in ~90% of cases).[42] One reason for this is paramedics lack access to alternatives.[44–46] There is a vision therefore of what could help: ambulance service access to some form of 'Alternative Care Pathway' (ACP) whereby those seeking help for an epileptic seizures judged not to require ED are cared for within less costly, alternative environments.

The exact nature of the ACP is not clear. Different regions and services are considering different configurations and doing so in an uncoordinated way. ACPs being considered appear to include paramedics transporting patients' home or to an urgent treatment centre rather than ED. Others involve paramedics leaving patients at scene with the offer of a telephone call from a nurse or general practitioner with a specialist interest in epilepsy within 24, 48 or 72 hours. A possible benefit of this latter configuration is health inequalities could be reduced by

introducing a mechanism by which all PWE 'in need' are brought to specialists' attention.

ACPs are not new[47] and paramedics have not been obliged to transport patients to ED since ~1997. The ambitions of ACPs have though increased and there is significant support within the UK policy realm to expand the number of conditions covered by them as a means of managing demands on hospital services.[48–50]

ACPs within the ambulance have largely come about due to the 'it seemed like a good idea at the time' principle,[51] rather than with reference to a behaviour change theory. Evidence on the utility of ACPs is though generally positive.[47 52–54] In their review of potential revisions to the urgent and emergency care system, the Nuffield Trust identified greater ambulance/paramedic triage in the community as having the most the positive evidence of effectiveness.[40] Paramedics appear willing and safely able to use ACPs when trained and for some ACPs there is evidence of cost-savings and greater patient satisfaction.[52 53]

In the case of epilepsy, qualitative research[44–46] provides the beginnings of a theoretical basis for the use of an ACP in epilepsy with the mechanisms by which it could make a difference being that it may: increase awareness and likelihood that paramedics will consider non-conveyance and referral pathways as an option in appropriate cases; increase paramedics' clinical knowledge of how to make appropriate non-conveyance decisions; increase paramedics' knowledge of alternative care providers that are acceptable to service users; and increase paramedics' confidence about making a non-conveyance decision and reducing anxiety about risk.

### Developing an ACP for epilepsy

An ACP for epilepsy holds potential. As a team of healthcare professionals, researchers and service user representatives with expertise in epilepsy and urgent care, our ultimate goal is therefore to evaluate the most promising ACP and use the evidence to transform service organisation nationally. However, we cannot currently do this because the way in which ACPs are being developed means it is not known which of the ACPs has the potential to be most effective and could be justified for selection.

Specifically, ACPs are being developed largely 'top-down', with the patient voice being absent. The nature and content of ACPs may not therefore align with what patients/carers would consider to meet their needs. Evidence suggests decisions to access healthcare services can be informed by how a patient/carer perceives their situation[55–57] and there can be a mismatch between patients/carers and health professionals' views, including of what constitutes an 'emergency'.[58–61] The upshot is the acceptability—a fundamental criterion an intervention needs to satisfy to be positioned to achieve its intended outcome[51]—of the different ACPs to patients and carers is unknown.

To date, only one epilepsy ACP has been reported on.[62] Despite the evaluation revealing positive outcomes, less than 10% of eligible PWE attended to by paramedics were put onto the ACP. The reasons for this low uptake rate were not explored. A possible explanation is that the ACP was not acceptable to PWE.

It has been assumed the target population *does not* want to be conveyed to ED and will readily accept an ACP. It may not be that straightforward. Some PWE certainly express dissatisfaction with being taken to ED.[63] Others though express a need for immediate access to urgent care; with some PWE and their family and friends (to whom care decisions are often delegated to when the patient is unconscious or lacks capacity) being fearful of seizures, including the possibility of brain damage.[25 26]

## CURRENT PROJECT

This 28 month project seeks to identify the ACP that should be prioritised for testing/implementation. It seeks to shape the change that results from the identified momentum so the likelihood of patients and the NHS benefiting are maximised and finite health resources used in an informed, rational way. It will bring patients/carers from the target population, healthcare professionals and commissioners together to identify which ACP configuration best encapsulates the features important to patients and carers *and* is feasible within the NHS context. The focus will be on the care of adults with epilepsy, rather than children/young people, since discussion to date relating to the use of ACPs and ED care has largely focused on the former (eg,.[19 62]). As per NICE guidelines for epilepsy,[39] adults are defined as those aged 18 years and older.

The project has three linked stages and will use survey techniques, qualitative methods, consensus meetings and Discrete Choice Experiments (DCEs) to achieve its aims (figure 1).[64 65] DCEs are an attribute-based survey methodology, underpinned by the theory that any 'good' (including a health service) can be described by its constituent characteristics (attributes), and that the extent to which an individual prefers a 'good' depends on the levels these attributes take.[66] We considered DCEs to provide an efficient, scientifically defensible, and non-tokenistic way of bringing the patient voice into ACP design since they allow a person's stated preferences and priorities to be elicited by presenting them with a hypothetical scenario (eg, having a seizure at home) and asking them to choose which of two (or more) care options described by a series of attributes they prefer. The process is then repeated with alternative care choices being presented. By varying attribute levels and observing how participants change responses, the importance of attributes and the extent to which they drive preference can be inferred.

### Aims

a. Systematically identify ACPs being considered by the NHS for epilepsy and describe extent to which service users have been involved in their design (Stage 1a).
b. Understand patient and carers' decision-making processes for seeking or not seeking ED care, and their concerns and expectations regarding ACPs (Stage 1b).

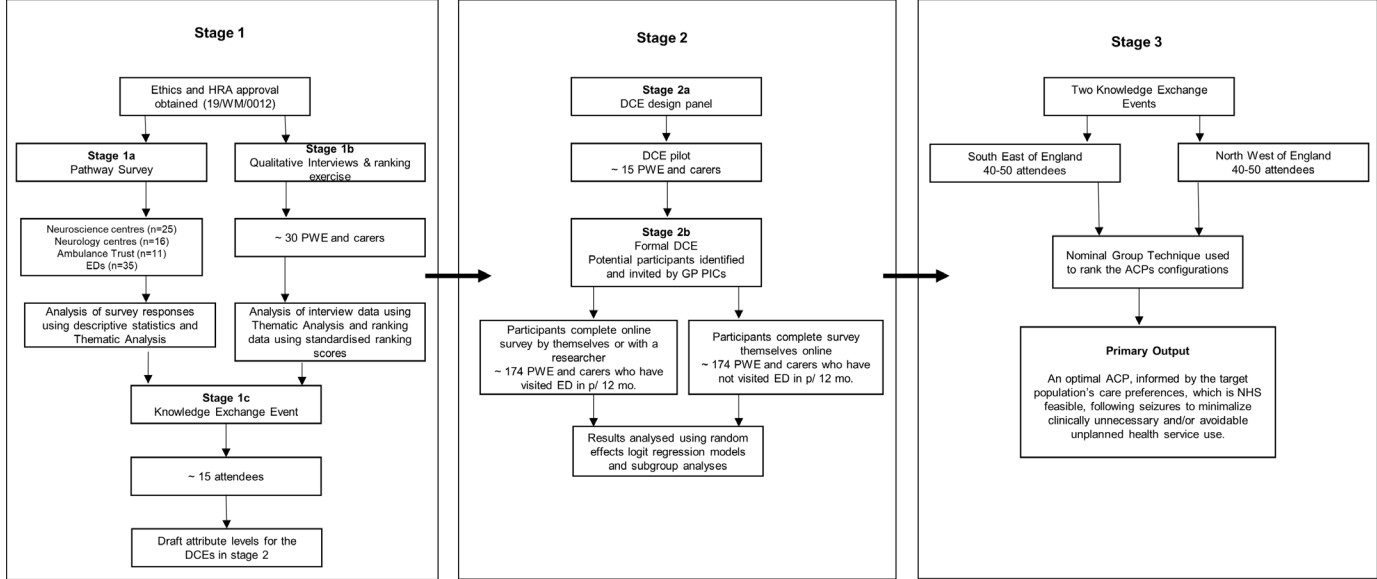

**Figure 1** Research project process diagram. ACP/s, alternative care pathway/s; DCEs, Discrete Choice Experiments; EDS, Emergency Departments; GP, General Practitioner; HRA, health research authority; NHS, National Health Service; PICs, Participant Identification Centres; PWE, People with Epilepsy.

c. Identify the attributes of post-seizure emergency care that PWE and their carers consider important and identify if this differs by the context in which the seizure occurs (Stage 1b).

d. Determine patient and carer preferences for post-seizure emergency care using DCEs and identify any subgroup differences (Stage 2b).

e. Estimate uptake of different ACPs configurations by patients and subgroup differences (Stage 2b).

f. Hold expert workshops to develop an optimal ACP following seizures, to be prioritised for implementation and evaluation via a subsequent project (Stage 3).

## METHODS AND ANALYSIS

This project will use qualitative and quantitative methods (figure 1). According to Greene *et al's*[67] conceptual framework, the purpose of using a mixed methods approach is 'development' (seeking to use the results from one method to help develop or inform the other method, where development is broadly construed to include sampling and implementation, as well as measurement decisions; Stages 1–2b) and 'expansion' (seeking to extend the breadth and range of inquiry by using different methods for different inquiry components; Stages 2b and 3).

Data collection and analysis for Stages 1a and 1b will be completed concurrently and independent from one another. From Stage 1c onwards, the design will be sequential and dependent. During Stage 2a, the findings from the earlier stages will be integrated to develop the DCE. Following completion of analysis for Stage 3, findings from all the stages will be integrated in a narrative form when interpreting and reporting on the project's findings. Particular attention will be given to how the findings from Stage 3 converge and diverge from the qualitative and quantitative findings from the earlier stages.

## Stage 1
### Stage 1a: pathway survey
*Purpose*
Systematically identify ACPs being considered. Will allow ACPs presented within DCEs to include main ones being considered.

*Design*
Cross-sectional survey of NHS service providers asking respondents whether their service has considered (or implemented) any changes to the management of those with epilepsy/suspected seizures to minimise clinically unnecessary/avoidable ED visits. Respondents will report how change is anticipated to minimise visits and whether service users informed change.

The survey will take ~10 min, be piloted and completed online. Survey respondents will be able to indicate if they want to be informed of subsequent project parts.

*Recruitment*
Invitations to participate, followed by two reminders, will be sent to clinical leads/directors of England's regional ambulance trusts (n=11), regional neuroscience (~n=25)[68] and neurology centres (~n=16),[69 70] and a random sample of 25% of its 'Type 1' EDs[71] (n=35), stratified by geographical area and size (attendances).

*Analysis*
Characteristics of responding and non-responding organisations will be compared. To determine the types of ACPs being considered and proportion of organisations considering each, a subset of surveys will be selected. A qualitative researcher, supported by the wider team, will read these to identify recurrent ACP types and collate them into a thematic coding framework. The framework will

be applied to the full data set and modified to ensure all ACPs types are captured.

### Stage 1b: qualitative interviews
#### Purpose
Understand patient and carer decision-making, the important attributes of post-seizure care, and concerns and expectations regarding ACPs. Will permit attribute development to adhere to good practice.[72 73]

#### Design
Face-to-face semi-structured interviews will be conducted with PWE, and where possible an informal carer, and shaped by a piloted topic guide (online supplementary file 1).

Participants will be asked for contextual information about their epilepsy and ED visits. The researcher will explore patients and carers confidence in managing seizures and the decision to call the ambulance services. The influences on ambulance use and ED use when seizures occur in public places will also be examined. Evaluative information will then be sought regarding positive and negative perceptions of the ambulance service and casualty as well as any ACPs experienced. Potential attributes will be identified by describing two known ACPs (ie, urgent care centres and being taken home) and asking participants for their expectations and concerns.

To help reduce the number of care attributes and improve face validity, participants will rank what they consider to be the five most important care attributes. Interviewer will help recall those mentioned and write them onto show cards.

To promote consistency, interviews will be conducted by one experienced qualitative researcher (AMK). AMK (PhD), is a university-based qualitative researcher with an interest in health services research but no specialist knowledge of, or involvement in, the ambulance or emergency medical services. Before agreeing to take part, participants will be given a participant information sheet detailing the research. Written informed consent will be obtained (by AMK) before the interviews. No non-participants will be present during the interviews. Interviews will take place at a time and place convenient to the participant. It is anticipated they will last 60–90 min.

To promote transparency, meticulous records of the interviews will be kept, interviews audio-taped, transcribed verbatim[74] and reported according to Consolidated criteria for Reporting Qualitative research.[75]

#### Recruitment
Aim is to capture a comprehensive range of perspectives, and sample size will be determined by data saturation. We anticipate[72 76] recruiting a purposive sample of ~30 PWE, with or without an informal carer. Table 1 lists the inclusion and exclusion criteria.

Eligible PWE will be identified by two means. First, persons with uncontrolled epilepsy who participated in a recent National Institute for Health Research (NIHR) funded 'Self-Management Education for adults with Epilepsy' trial in the South-East of England and expressed interest in future projects will sent notification of the study.[77] Second, adverts will be circulated to persons affiliated with epilepsy user groups, including Epilepsy Action. Persons interested in taking part will be asked to contact the research team. To identify carer participants, patients will be asked to choose a significant other/informal carer to take part. Participants will receive a £20 shopping voucher.

#### Analysis
Analysis will proceed alongside data collection and be based on a Framework approach.[78] This approach ensures each case is fully accounted for and is regarded as particularly helpful for addressing policy questions. The primary aim of analysis will be to generate an exhaustive list and description of the salient attributes of care (and levels) identified by participants.

The analytic approach will involve the identification of initial themes that are then grouped into main themes and subthemes. Transcripts will be read and re-read, and coded. At least two members of the research team (AMK, MM, LR) will participate in all data analysis to reduce bias in the identification and interpretation of themes.

A thematic 'chart' will be created to summarise information for each theme to allow cross-case and within-case analysis through a process of constant comparison, with particular attention being paid to deviant cases. Analysis will pay attention to the similarity and differences in salient attributes according to where the seizure occurred (home, public), who was there, who made the 999 call, whether the patient had seen a specialist in the prior 12 months, and whether they self-reported an intellectual impairment. Participant quotations (with minor editing where necessary to preserve anonymity) will be presented to illustrate themes and help verify interpretation. QSR International's NVivo 11 qualitative data analysis software[79] will be used as a management tool throughout the process.

The attributes nominated during the ranking exercise will be categorised according to the thematic framework, then analysed quantitatively (by EAH) using standardised rank scores.

### Stage 1c: knowledge exchange event
#### Purpose
It is important the DCE elicits views on ACPs that could plausibly be delivered within the NHS. This event will therefore, early in the project, determine feasibility of the attributes and associated levels that have been identified by service users in Stage 1b.

#### Design
One-day event at which healthcare professional representatives will be shown draft attributes, levels, and choice sets and asked to deliberate each choice. For instance, should Stage 1b indicate service users consistently say they prefer to be transported home by the ambulance

**Table 1**  Participant inclusion and exclusion criteria

| Study part | Inclusion criteria | Exclusion criteria |
|---|---|---|
| **Stage 1b—Qualitative interviews** | | |
| | ► Established diagnosis of epilepsy, or an informal carer for someone with epilepsy<br>► Age≥18 years (no upper age limit)<br>► Have visited ED in the past 12 months for epilepsy (as reported by the patient)*<br>► Able to provide informed consent and communicate in English | ► Severe current psychiatric disorders (eg, acute psychosis)<br>► Life-threatening medical illness |
| **Stage 2—Discrete choice experiments** | | |
| Group 1<br>Patients | ► Clinically confirmed diagnosis of epilepsy (for any duration)<br>► Any epilepsy syndrome and any types of focal or generalised seizures<br>► Currently being prescribed antiepileptic medication<br>► Age≥18 years (no upper age limit)<br>► Have visited ED in the past 12 months for epilepsy<br>► Able to provide informed consent and independently complete a questionnaire in English<br>► Lives in the North West of England | ► Severe current psychiatric disorders (eg, acute psychosis)<br>► Life-threatening medical illness<br>► Resides within a care or nursing home or of no fixed abode |
| Carers | ► A significant other to a person with epilepsy (eg, family member, friend) who the patient identifies as providing informal support or self-identifies themselves if the patient has a substantial intellectual disability<br>► The person with epilepsy they care for has visited ED in the past 12 months<br>► Age≥16 years (no upper age limit)<br>► Able to provide informed consent and independently complete a questionnaire in English<br>► Lives in the North West of England | ► Severe current psychiatric disorders (eg, acute psychosis)<br>► Life-threatening medical illness |
| Group 2<br>Patients | ► Clinically confirmed diagnosis of epilepsy (for any duration)<br>► Any epilepsy syndrome and any types of focal or generalised seizures<br>► Currently being prescribed antiepileptic medication<br>► Age≥18 years (no upper age limit)<br>► Has *not* visited ED in the past 12 months, but has had a seizure with loss of awareness in this period<br>► Able to provide informed consent and independently complete a questionnaire in English<br>► Lives in the North West of England | ► Severe current psychiatric disorders (eg, acute psychosis)<br>► Life-threatening medical illness<br>► Resides within a care or nursing home or has no fixed abode |
| Carers | ► A significant other to a person with epilepsy (eg, family member, friend) who the patient identifies as providing informal support or self-identifies themselves if the patient has a substantial intellectual disability<br>► The person with epilepsy they care for has *not* visited ED in the past 12 months, but had a seizure with loss of awareness in this period<br>► Age≥16 years (no upper age limit)<br>► Able to provide informed consent and independently complete a questionnaire in English<br>► Lives in the North West of England | ► Severe current psychiatric disorders (eg, acute psychosis<br>► Life-threatening medical illness |

*Should recruitment prove slower than anticipated, the recruitment criteria will be relaxed to allow people who have had ED or ambulance contact within the last 2 years.
ED, emergency department.

service regardless of distance, then representatives would be asked to what constitutes a feasible distance. Our team has previously used this approach.[80]

### Recruitment

Around n=15 ambulance staff and neurology service representatives. Persons will be recruited from organisations participating in Stage 1a with the help of a sampling framework to ensure representatives come from geographically diverse areas. Representatives will be reimbursed for travel. Those not able to attend will be able to provide feedback remotely.

### Analysis

With attendees' consent, the session will be audio-recorded and field notes taken. Feedback will be used to optimise the attribute levels within the DCEs.

## Stage 2

### Stage 2a: developing the DCE
### Purpose

Generate DCE questionnaire to elicit patient and carer preferences for post-seizure care and which meets best practice guidance.[72 81]

### Design

A multidisciplinary panel, chaired by DAH, will compose a draft DCE questionnaire using Stage 1 evidence and refine it on the basis of a pilot. The panel will include expertise in DCEs (EAH), emergency medicine (SG), neurology (AGM), paramedical science (MJ), general practice (JMD) and psychology (AJN), as well as having service user representation.

The panel will form a consensus on which restricted number of ACPs best represent the range being considered and are to be valued using the DCE. Using the list of candidate attributes generated from Stage 1b the panel will select 4–6 attributes by which to describe the packages.

In selecting attributes, the panel will favour those prioritised by service users, which are modifiable, capable of being traded and best describe the ACPs.[72] Attributes deemed to be too close to the latent construct of the utility derived from the ACP (eg, 'satisfaction') will not be selected.

DCEs will be generated for the three most common presentations to ED by PWE. For each, participants will be presented with a seizure scenario vignette and via standardised instructions complete a series of forced, pairwise comparisons to indicate which care packages they would prefer. Vignettes will be developed by the panel based on clinical and lived experience. Online supplementary file 2 provides an illustration of what a DCE *might* look like.

Plausible attribute levels will be specified on the basis of findings from Stage 1, expert opinion and relevant literature. A strength of DCEs is the possibility of varying attribute levels to also model preferences within future contexts. Using its expertise and knowledge, the panel

will seek to assign levels to attributes to account for major anticipated changes—such as longer ED waiting times.[82]

A fractional factorial design will identify a purposeful subset of options for each DCE, to allow an estimate of the main effect of each attribute independently and selected two-factor interactions, while minimising the number of paired comparisons participants are asked to make. The profiles will be obtained from a published design catalogue.[83] Binary choices will then be created using the fold-over method, which replaces each attribute level systematically.[84]

Once drafted, the DCE will undergo iterative individual pilot testing with ~15 PWEs and carers from the project's Patient and Public Involvement (PPI) group to check clarity and duration. Pilotees will 'think aloud'[85] when making choices, and asked to consider their preferred presentation of attributes (eg, text, pictograms). Sessions will be audio recorded, notes made and refinements made.

### Stage 2b: formal DCE
### Purpose

Determine patient and carer preferences for post-seizure emergency care, estimate uptake of different ACPs configurations, and subgroup differences.

### Design

A representative sample of PWE and informal carers who have visited ED in the last 12 months, as well as those at risk, will be sought to complete the finalised DCE questionnaire. PWE who have uncontrolled epilepsy, but who have not visited ED in the last 12 months will be asked to visit an online survey page and complete the DCE questionnaire using their own internet enabled devices. Due to potentially greater social disadvantage, patients who *have* visited ED in the prior 12 months will have the additional option of completing the DCE during a face-to-face appointment with a researcher who will provide an internet enabled device.

To limit burden, participants will complete DCEs relating to just two seizure scenarios (allocated at random). As well as the DCEs, respondents will report on their demographics, epilepsy, knowledge and fear of seizures and service use. Each participant will receive a £20 shopping voucher.

### Recruitment

General Practices in North-West England will identify potential patient participants by searching their epilepsy registers.[86] Practices will be recruited with the help of the NIHR Clinical Research Network North West Coast. Practices will send ostensibly eligible patients invitation packs via 'Docmail' and a repeat invite 2 weeks later to those not responding. Patients not wishing to participate will be encouraged to notify the team and detail their reasons. Eligibility criteria are in table 1.

A definitive sample size calculation depends on the finalised design of the DCE.[87] To permit decisions regarding recruitment to be made in the meantime

we anticipated the design and calculated the required sample size by using Orme's[88] formula. Full details are provided in box 1. This indicated 348 PWE (or associated informal carers) would be required, comprising data 174 patients (or associated carers) who have visited ED in the prior 12 months and 174 patients (or associated carers) who have not.

**Formula used and factors considered in estimating likely sample size required**

► Orme's formulae: $(n*t*a)/c$ must be ≥500, where $n$ is number of respondents, $t$ number of tasks, $a$ number of alternatives per task, and $c$ number of analysis cells (for main effects, the largest number of levels for any one attribute).
► Anticipated discrete choice experiments (DCEs) to involve participants being asked to make ($t$) nine separate choices between ($a$) two care packages and ($c$) three would be maximum attribute levels. Thus, Orme's formula indicated complete data from n=84 needed.
► Intend to complete DCEs for three different seizure scenarios, but participants would only complete two, therefore 126 participants required.
► To allow for variation in the finalised number of choice tasks and levels we increased this to 150 and further inflated it to 174 to accommodate for ~15% of missing data experienced by prior epilepsy DCEs.
► As an objective was to describe and compare care preferences of two subgroups (patients (and associated carers) who have visited an ED in the prior 12 months versus patients (and associated carers) who have had a seizure in the prior 12 months, but have not visited an ED), 174 patients (and associated carers) who have visited ED in the prior 12 months and 174 patients (and associated carers) who have not, will need to be recruited.
► Total sample size, thus n=348.

Based on responses to prior health DCEs[89] and studies with PWE,[16 24] we anticipate 30%–60% of those invited will agree to participate. An average English GP practice[90] will have 24 PWE aged ≥18 on their register who have experienced ≥1 seizure in the prior 12 months,[9] with 10 having visited an ED during that period.[12–14] Governed by the need to recruit a subgroup of 174 PWE who have visited ED in the prior 12 months, 29–58 general practices will be required.

### Analysis

Data for the different seizure vignettes will be analysed separately using an error components logit model that allows for a panel of responses from the same respondent. This will determine the importance of the different care attributes for preference and direction of effect (as indicated by the significance of the coefficients and their size). Comparison between vignettes will depend on attribute and level specification of each experiment. Analyses will be completed with and without respondents failing a 'rationality' test.

Marginal rates of substitution (the rate at which respondents are willing to give up a unit change in one attribute in exchange for a unit change in another while maintaining the same level of utility) will be calculated using each quantitative attribute as the value attribute with bootstrapped confidence intervals (1000 reps).

Preference heterogeneity will be assessed by estimating a mixed logit model, which will include random parameters for the attributes, and interactions between attributes and respondent characteristics.[91] A-priori characteristics of interest are: (1) emergency care for epilepsy in the past 12 months (paramedic and/or ED visit), (2) social deprivation (with participants grouped according to how socially deprived the area is within which they live); (3) whether or not they have seen an epilepsy specialist within the prior 12 months (which they should have according to NICE guidelines[88] given they will all have uncontrolled epilepsy); and (4) whether they or the person they know in the case of carers, self-reports an intellectual disability.

'Utility scores' for different ACP configurations and significantly different populations will be calculated and ranked accordingly. Their uptake under different seizure scenarios will be estimated.

All analyses will be conducted in STATA V.11.

### Stage 3: knowledge Exchange workshops
#### Purpose
Disseminate Stage 1 and 2 findings to those positioned to develop, fund, support and run ACPs and use their expertise to identify which ACP/s favoured by patients and carers is most NHS feasible and should be prioritised for implementation/evaluation.

#### Design
One Knowledge Exchange event in South England, one in the North. They will start with presentation of project's findings, including ACP configurations ranked according to service user preference. A Nominal Group Technique, described by the James Lind Alliance,[92] will then be used to hear delegates' views of the ACPs and rank them according to the extent to which they meet users' preferences and are NHS feasible. Facilitators will encourage delegates to consider barriers to change, supply constraints, acceptability of the service to staff, possible cost and potential to redress healthcare inequalities.

Voting will be completed using electronic devices. With attendees' consent, sessions will be audio-recorded, and a qualitative researcher will observe and record field notes.

#### Recruitment
Aim to have 40–50 health service managers, commissioners, healthcare professionals and service user representatives at each event. Persons will be recruited mainly from the institutions participating in Stage 1a. Additional attendees will come from commissioning groups and the epilepsy patient groups. The project's PPI group will also be invited. A sampling matrix will be formed to help ensure broadly equal representation of persons from health professional, commissioning and managerial roles.

## Analysis

The rank order of the ACPs that results from delegates voting will be described. Depending on findings from the previous stages, the ACPs might have been presented to delegates according to seizure scenario. If so, rank orders for the ACPs for the different scenarios will be determined and compared. As a secondary objective, rank orders for the different representative subgroups will be determined. Transcripts from the events will be thematically analysed to capture delegates' views and justifications for their preferences.

## DISCUSSION

### Patient and public involvement

This research was driven by a need to consider the views of PWE when designing an ACP for epilepsy. Epilepsy Action, the largest user organisation in the UK, contributed to the design of the research, and facilitated a PPI event with 23 service users to discuss project design and conduct. A PPI group will be established, including PWE and carers. They will be supported by Epilepsy Action who have an active PPI scheme and reimbursed for travel and time according to INVOLVE.[93] The group will contribute as research peers by: providing feedback on research materials; interpretation of findings; piloting the DCE; and being delegates in workshops. The group will be encouraged to help with dissemination, including presenting the findings at the Stage events and teaching.

### Ethics and dissemination

Monitoring by an independent Study Steering Committee (SSC) will help to ensure the rights, safety and well-being of participants are the most important considerations. Only persons providing informed consent will participate. The SSC will be composed according to our funder guidelines.[94] Compliance with the principles of Good Clinical Practice and scientific integrity will be managed by the study management team through regular and ad hoc meetings.

The project will identify what, if any, ACP configuration/s is most acceptable to service users and has the backing of those expected to deliver and support it. To ensure maximum impact those considering an ACP need to be aware of the project, have confidence in its outputs and use its findings. To help ensure this: (1) Representatives from all of the relevant service providers in the UK will be informed of the project (including notifying the leads of England's Sustainability and Transformation Partnerships and Urgent and Emergency Care Vanguards); (2) professional bodies will announce its funding (Acknowledgements); (3) the research team includes appropriate expertise and opinion leaders; and (4) representatives from relevant NHS services will be invited to take part in Stage 1a and 3a.

The project's findings will be disseminated via international conferences and at least four journal manuscripts. The manuscripts will be published open access and non-technical reports will be circulated via NHS network newsletters and Epilepsy Action. Participants will be provided with a copy of the results.

Having identified the ACP, funding will stop but the team would, via a subsequent project seek to implement and evaluate the chosen ACP. The nature of that evaluation will be informed by the state of the NHS closer to the time. The Medical Research Council[95] highlight intervention evaluation takes place in a wide range of settings and many factors can constrain choice of evaluation methods. One trade-off that may need consideration is time until results become available/demand versus methodological rigour.

**Author affiliations**
[1]Department of Health Services Research, University of Liverpool, Liverpool, UK
[2]Department of Basic and Clinical Neuroscience, Institute of Psychiatry, Psychology & Neuroscience, King's College London, London, UK
[3]Centre for Health Economics & Medicines Evaluation, Bangor University, Bangor, UK
[4]Institute of Pharmaceutical Science, King's College London, London, UK
[5]Basic & Clinical Neuroscience, King's College London, London, UK
[6]Academic Unit of Primary Medical Care, The University of Sheffield, Sheffield, UK
[7]North West Ambulance Service NHS Trust, Bolton, UK
[8]Centre for Health Economics and Medicines Evaluation, Bangor University, Bangor, UK
[9]Medical Care Research Unit, University of Sheffield, Sheffield, UK
[10]Department of Molecular and Clinical Pharmacology, University of Liverpool, Liverpool, UK

**Acknowledgements** The authors would like to thank Epilepsy Action (Angie Pullen, Amanda Stoneman) for the role that they will have in this project. They would like to express appreciation for the contributions from people with epilepsy and their family members and friends who will participate in this study and thank members of the Study Steering Committee (Tom Quinn (chair), Verity Watson, Trevor Baldwin, Nigel Rees, Khalid Hamandi, Daniel Horner, Juliet Bransgrove, and Jayne and Sam Burton). We also acknowledge the International League Against Epilepsy (British Branch), Royal College of Emergency Medicine and the Epilepsy Nurses Association for informing members of the project. The study sponsor is the University of Liverpool (reference: UoL4258; sponsor@liv.ac.uk).

**Contributors** AJN conceived of the study and designed it together with EAH, DAH, LR, MM, MJ, JMD, SG and AGM. EAH planned the statistical analysis for the DCE. AJN and AM wrote the manuscript, with revisions being made by EAH, AM, DAH, LR, MM, MJ, JMD, SG and AGM. All authors read and approved the final manuscript.

**Funding** This project is funded by the National Institute for Health Research's Health Services and Delivery Research Programme (HS&DR Programme) (project number 17/05/62). The views and opinions expressed herein are those of the authors and do not necessarily reflect those of the University of Liverpool, the HS&DR programme, the NIHR, the NHS, or the Department of Health and Social Care.

**Competing interests** None declared.

**Patient consent for publication** Not required.

**Ethics approval** This study has received Health Research Authority approval and favourable ethical opinions from NRES Committee West Midlands–Solihull (19/WM/0012) and King's College London's Psychiatry, Nursing and Midwifery Ethics Committee (LRS-18/19-10353).

**Provenance and peer review** Not commissioned; externally peer reviewed.

**Data availability statement** There are no data in this work.

**ORCID iDs**
Adam J Noble http://orcid.org/0000-0002-8070-4352
Alison McKinlay http://orcid.org/0000-0002-3271-3502
Jon Mark Dickson http://orcid.org/0000-0002-1361-2714

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
