## [Reviewer comments · BMJ Open]

ARTICLE DETAILS

TITLE (PROVISIONAL)	Developing patient-centred, feasible alternative care for adult emergency department users with epilepsy: Protocol for the mixed-methods observational 'Collaborate' project.
AUTHORS	Noble, Adam; Mathieson, Amy; Ridsdale, Leone; Holmes, EA; Morgan, Myfanwy; McKinlay, Alison; Dickson, Jon; Jackson, Mike; Hughes, Dyfrig; Goodacre, Steve; Marson, Anthony

VERSION 1 – REVIEW

REVIEWER	Anthony Fine Assistant Professor of Neurology Mayo Clinic USA
REVIEW RETURNED	04-Jun-2019

GENERAL COMMENTS	The authors present a project aimed at creation/identification of an alternative care pathway for patients with epilepsy to reduce unnecessary and inappropriate health services. This is an ambitious project which will depend heavily on stakeholder participation and recruitment of patients with epilepsy and their carers. The inclusion criteria may need to be less stringent when recruiting individuals who have not had an emergency department visit or ambulance contact within the past 12 months in order to improve recruitment. The authors provide a practical multi-staged approach to this project. I think it would be worthwhile to publish this project.
--

REVIEWER	Wanqing Zhang UNC Chapel Hill USA
REVIEW RETURNED	14-Jun-2019

GENERAL COMMENTS	The ACP proposed in this protocol could be a practical alternative for PWE instead of using ED. See my comments below: 1. This protocol does not include a conceptual model/framework guiding the study plan. It is not clear what the mechanisms might be that would explain unmet need PWD visiting ED.2. Detailed description is needed in terms of quantitative and qualitative data analysis. Please describe a systematic approach for qualitative data analysis, including planned validation process (eg, How the researchers validate the qualitative data with the interview participants? Will the transcripts read by the participants to ensure intent is accurate?)3. Clarify if a sequential mixed design or parallel mixed design will be used. This mixed methods project does not include a plan on
--

	how to integrate qualitative and quantitative findings. For example, quantitative and qualitative results can be integrated in both results and interpretation phase. The integration focuses on how the results from both methods are similar or different, with the primary purpose being to support each other. Qualitative data can provide a different perspective from those not included in the quantitative survey/instrument. 4. The dissemination plan is rather general and not specific to the study objective.
--	---

REVIEWER	Elizabeth Cecil Imperial College London, UK
REVIEW RETURNED	24-Jun-2019

GENERAL COMMENTS	I think this is a mostly well written protocol outline and a very important piece of work Below are my comments Introduction Not all sub-headings make sense. For example the sub-heading 'Unmet need PWE visiting ED'. The last two paragraphs in in this sub-heading need to be clearer. Aims The aims do not obviously link to project stages. Can this be made clearer? Figure 1 is important but does not include all stages. Can you add these? Methods Stage 1b Design: Informed consent? Signed consent forms are mentioned later but what about information sheets? Who will conduct the semi-structured interviews? How long will the interviews be? Stage 2b Recruitment: General practices are recruited from one region, will this mean that there is an issue with generalisability? Analysis: Can you provide a bit more information on outcome and explanatory variables for the random effects logistic regression models. Table 1 Not clear why patients need to be 18+ but carers 16+ Are views of children with epilepsy (or their carers) represented? Group 2 carers the first bullet point lives with the patient group. Overall there are a lot of acronyms. A list of acronyms may help. Can Epilepsy Action be in full? Initials for research team members could be in brackets to differentiate from acronyms. Citations should be after punctuation but appears in text both before and after.
--

VERSION 1 – AUTHOR RESPONSE

REVIEWER 1

REVIEWER COMMENT 1.1: "...This is an ambitious project which will depend heavily on stakeholder participation and recruitment of patients with epilepsy and their carers. The inclusion criteria may need to be less stringent when recruiting individuals who have not had an emergency department visit

or ambulance contact within the past 12 months in order to improve recruitment. The authors provide a practical multi-staged approach to this project. I think it would be worthwhile to publish this project.”

AUTHOR RESPONSE: We thank the reviewer for highlighting the importance of the project and for their suggestion. We will review the inclusion criteria if recruitment requires improvement.

REVIEWER 2

REVIEWER COMMENT 2.1: “...It is not clear what the mechanisms might be that would explain unmet need PWE visiting ED.”

AUTHOR RESPONSE: In view of the reviewer’s comment have now given greater detail within the manuscript to this issue. Specifically, on page 4 of the revised manuscript, under the section titled “Unmet needs of PWE visiting ED” we now note the following:

“Unmet needs of PWE visiting ED

Whilst the acute episodes leading PWE to visit ED do not typically require emergency care, the visits can be expressions of the person having received suboptimal ambulatory care and unmet needs. attendees do often require improvements in ambulatory care to prevent re-attendance. NASH (24), for instance, found that most

(~65%) PWE visiting ED are not known to specialist epilepsy services and many were via their usual care provider, seemingly receiving outdated care. For instance, despite focal epilepsy being the most common epilepsy type, most likely to be refractory and often not best treated with the medication sodium valproate, NASH found it to be the most prescribed medication amongst ED attendees. Coping with life in the context of epilepsy also requires an individual to accept their diagnosis and learn and adopt a range of self-management behaviours to prevent seizures and manage consequences. PWE visiting ED though and their significant others appear to have less knowledge about epilepsy and its management, including seizure first aid.(7, 25-27) They might therefore benefit from enhanced selfmanagement support, like that provided by epilepsy nurses. The reason that knowledge might be low in ED attendees is because there remains no routine course that all PWE can go on to learn about epilepsy (as there are for some conditions). People who have lower education levels appear to fare worst from this situation with them having been found to have the least epilepsy knowledge.(28, 29)

That some PWE in the UK are receiving suboptimal care is well known, with there being longstanding challenges in Ensuring that the PWE most in need of specialist care receive it has been a longstanding challenge. Whilst trial evidence indicates ~70% of PWE can become seizure free, only ~50% of PWE in the UK currently are,(30) with PWE in socially deprived areas faring the worst.(31) PossibleFactors contributing to the challenge are that reasons include the UK’s has a comparatively small specialist workforce small number of neurologists (32) and that there is no national, incentivised system to identify those within the epilepsy population that might need stepped-up care.the lack of incentives for the identification of such PWE. To further complicate matters, GPs, who care for most PWE, have expressed low confidence in managing the condition. An indication of the challenge is that trial evidence indicates ~70% of PWE can become seizure free, but only ~50% of PWE in the UK currently are.(30) PWE in socially deprived areas again fare worst.(31)

ED visits by PWE can be considered opportunities to intervene. Unfortunately, under current arrangements, going to ED does not typically lead to PWE receiving ambulatory care improvements. Bodies, such as NICE(2), recommend that when seizures are not controlled, a patient should be referred to specialist services (within 4 weeks) since this may improve outcomes, including rendering some seizure free.(33, 34) Going to ED does not currently lead to PWE typically receiving ambulatory care improvements. Most (80%) PWE visiting ED are though not seen by a specialist during their attendance, their usual care providers may not be informed of the attendance (13) and most (60%) PWE are not referred to a specialist for follow-up.

PWE living in the most deprived areas, as well as the elderly, are amongst the least likely to be referred on from ED.(35) The low referral rate may be due to some clinicians holding an incorrect nihilistic view that intractability is inevitable if seizure control is not obtained within a few years of therapy onset.

Assumptions about the willingness of certain patient groups to attend clinics may also be being made.(36, 37)”

In addition, under the section on page 6 titled “An ‘Alternative Care Pathway” we now note the following:

“...A possible benefit of this latter configuration is health inequalities could be reduced by introducing a mechanism by which all PWE ‘in need’ are being brought to specialists’ attention.

ACPs are not new (48) and paramedics have not been obliged to transport patients to ED since ~1997. The ambitions of ACPs have though increased and there is significant support within the UK policy realm to expand the number of conditions covered by them as a means of managing demands on hospital services

(e.g. (49-51)).

ACPs within the ambulance have largely come about due to the ‘ISLAGIATT’ (‘it seemed like a good idea at the time’) principle,(52) rather than with reference to a behaviour change theory. Evidence on the utility of ACPs is though generally positive.(48, 53-55) In their review of potential revisions to the urgent and emergency care system, the Nuffield Trust identified greater ambulance/paramedic triage in the community as having the most the positive evidence of effectiveness.(41) Paramedics appear willing and safely able to use ACPs when trained and for some ACPs there is evidence of cost-savings and greater patient satisfaction.(53, 54)

In the case of epilepsy, qualitative research (45-47) provides the beginnings of a theoretical basis for the use of an ACP in epilepsy with the mechanisms by which it could make a difference being that it may: increase awareness and likelihood that paramedics will consider non-conveyance and referral pathways as an option in appropriate cases; increase paramedics’ clinical knowledge of how to make appropriate nonconveyance decisions; increase paramedics’ knowledge of alternative care providers that are acceptable to service users; and increase paramedics’ confidence about making a non-conveyance decision and reducing anxiety about risk.”

REVIEWER COMMENT 2.2: “Detailed description is needed in terms of quantitative and qualitative data analysis. Please describe a systematic approach for qualitative data analysis, including planned validation process (eg, How the researchers validate the qualitative data with the interview participants? Will the transcripts read by the participants to ensure intent is accurate?)”

AUTHOR RESPONSE: In light of the reviewer’s suggestion, we have now added the following additional text describing the handling and analysis of the data arising from the qualitative interviews and the analysis of the DCE data. Participants will be not be provided with transcripts to ensure intent is accurate.

Text relating to the qualitative interview data:

Design: Face-to-face semi-structured interviews will be conducted with PWE, and where possible an informal carer, and shaped by a piloted topic guide (Supplementary File 1).

Participants will be asked for contextual information about their epilepsy and ED visits. The researcher will explore the decision-making surrounding patients and carers confidence in managing seizures and the decision to call the ambulance services. The influences on ambulance use and ED use when seizures occurring in public places will also be examined. seeking or not seeking ED care following recent seizures.Evaluative information will then be sought regarding by respondents describing positive and negative perceptions of the ambulance service emergency care and casualty

as well as any ACPs experienced. The researcher will explore the decision-making surrounding patients and carers seeking or not seeking ED care following recent seizures. Potential attributes will shall be identified obtained by describing two known ACPs (i.e., urgent care centres and being taken home) and asking participants for their expectations and concerns.

To help reduce the number of care attributes and improve face validity, participants will rank what they consider to be the five most important care attributes. Interviewer will help recall those mentioned and write them onto show cards.

To promote consistency, interviews will be conducted by one experienced qualitative researcher [AMK]. AMK (PhD), is a university-based qualitative researcher with an interest in health services research but no specialist knowledge of, or involvement in, the ambulance or emergency medical services. Before agreeing to take part, participants will be given a participant information sheet detailing the research. Written informed consent will be obtained [by AMK] before the interviews. No non-participants will be present during the interviews. Interviews will take place at a time and place convenient to the participant. It is anticipated they ; and will last 60-90 minutes.

To promote transparency, meticulous records of the interviews will be kept, interviews audio-taped, transcribed verbatim (75) and conducted and reported according to COREQ.(76)

And

“Analysis: Data collection and analyses will proceed in an iterative manner. An qualitative researcher, supported by the wider team, will analyse the data Analysis will proceed alongside data collection and be based on a Fframework approach.(79) This approach is suitable for smaller samples and ensures each case is fully accounted for and is regarded as particularly helpful for addressing policy questions. The primary aim of analysis will be to generate an exhaustive list and description of the salient attributes of care (and levels) identified by participants.

The analytic approach will involve the identification of initial themes that are then grouped into main themes and subthemes. Transcripts will be read and re-read, and coded line-by-line. At least two members of the research team (AMK, MM, LR) will participate in all data analysis to reduce bias in the identification and interpretation of themes.

A thematic ‘chart’ will be created to summarise information for each theme to allow cross-case and within-case analysis through a process of constant comparison, with particular attention being paid to deviant cases. Analysis will pay attention to the similarity and differences in salient attributes according to where the seizure occurred (home, public), who was there, who made the 999 call, whether the patient had seen a specialist in the prior 12 months, and whether they self-reported an intellectual impairment. Participant quotations (with minor editing where necessary to preserve anonymity) will be presented to illustrate themes and help verify interpretation.

NVivo 11 software will store data and support coding.(80) QSR International's NVivo 11 qualitative data analysis software (80) will be used as a management tool throughout the process.

thematically to generate an exhaustive list and description of the salient attributes of care (and levels)

identified by participants.

Transcripts will be read and re-read, and coded line-by-line.

The attributes nominated during the ranking exercise will be categorised according to the thematic framework, then analysed quantitatively [by EH] using standardised rank scores.

Quantitative (DCE):

“Analysis: Data for the different seizure vignettes will be analysed separately using random effects logit regression modelsthat allows for multiple observations from the same respondent. This will determine the importance of the different care attributes for preference and direction of effect (as indicated by the significance of the coefficients and their size). Comparison between vignettes will

depend on attribute and level specification of each experiment. Analyses will be completed with and without respondents failing a 'rationality' test.

Marginal rates of substitution (the rate at which respondents are willing to give up a unit change in one attribute in exchange for a unit change in another while maintaining the same level of utility) will be calculated using each quantitative attribute as the value attribute with bootstrapped confidence intervals (1000 reps). Analyses will be completed with and without respondents failing a 'rationality' test.

Heterogeneity in preference will be assessed, using subgroup analysis and log likelihood ratio testing, and the model specified accordingly.⁽⁹¹⁾ A-priori subgroups of interest are: (1) emergency care for epilepsy in the past 12-months (paramedic and/or ED visit), (2)

The influence of participants' characteristics on preference, including whether they had visited an ED in the prior 12 months and whether they were a patient or a carer, will be examined as a secondary analysis to allow us to determine what differences exist. We shall also conduct subgroup analyses to explore potential differences in preference based on: (1) social deprivation (with participants grouped according to how socially deprived the area is within which they live); (3) whether or not they have seen an epilepsy specialist within the prior 12 months (which they should have according to NICE guidelines [89] given they will all have uncontrolled epilepsy); and (4) whether they or the person they know in the case of carers, self-reports an intellectual disability. The influence of participants' characteristics on preference, including whether they had visited an ED in the prior 12 months, whether they were a patient or carer, social deprivation, and presence of an intellectual impairment, will be examined as a secondary analysis.

Marginal rates of substitution (the rate at which respondents are willing to give up a unit change in one attribute in exchange for a unit change in another while maintaining the same level of utility) will be calculated using each quantitative attribute as the value attribute with bootstrapped confidence intervals (1000 reps).. Comparison between vignettes will depend on attribute and level specification of each experiment.

'Utility scores' for different ACP configurations and significantly different populations will be calculated and ranked accordingly. Their uptake under different seizure scenarios will be estimated.

All analyses will be conducted in STATA 11 (StataCorp, College Station, TX, USA).

).

REVIEWER COMMENT 2.3: "Clarify if a sequential mixed design or parallel mixed design will be used. This mixed methods project does not include a plan on how to integrate qualitative and quantitative findings. For example, quantitative and qualitative results can be integrated in both results and interpretation phase. The integration focuses on how the results from both methods are similar or different, with the primary purpose being to support each other. Qualitative data can provide a different perspective from those not included in the quantitative survey/instrument."

AUTHOR RESPONSE: In light of the reviewer's comment, we now note the following within the manuscript (page 9):

"METHODS AND ANALYSIS

This project will use qualitative and quantitative methods (see Figure 1). According to Greene et al.'s (68) conceptual framework, the purpose of using a mixed methods approach is 'development' (seeking to use the results from one method to help develop or inform the other method, where development is broadly construed to include sampling and implementation, as well as measurement decisions; Stages 1-2b) and 'expansion' (seeking to extend the breadth and range of inquiry by using different methods for different inquiry components; Stages 2b and 3).

Data collection and analysis for Stages 1a and 1b will be completed concurrently and independent from one another. From Stage 1c onwards, the design will be sequential and dependent. During Stage 2a, the findings from the earlier stages will be integrated to develop the DCE. Following

completion of analysis for Stage 3, findings from all the stages will be integrated in a narrative form when interpreting and reporting on the project's findings. Particular attention will be given to how the findings from Stage 3 converge and diverge from the qualitative and quantitative findings from the earlier stages."

REVIEWER COMMENT 2.4: "The dissemination plan is rather general and not specific to the study objective."

RESPONSE: Whilst we know that a more detailed dissemination plan could be useful, we politely decline to develop this section further so as to keep the article (having addressed the other points from the reviewers) broadly in line with the recommended word limit for protocols.

REVIEWER 3

REVIEWER COMMENT 3.1: "Introduction - Not all sub-headings make sense. For example the sub-heading 'Unmet need PWE visiting ED'. The last two paragraphs in this sub-heading need to be clearer.

AUTHOR RESPONSE: We have now clarified the sub-headings (please see pages 4 & 7). We have also revised the last two paragraphs in the 'Introduction' section (please see page 8).

REVIEWER COMMENT 3.2: "The aims do not obviously link to project stages. Can this be made clearer?"

AUTHOR RESPONSE: Thank you for this suggestion. We have added which stage of the project each of the aims relate to on page 9.

REVIEWER COMMENT 3.3: "Figure 1 is important but does not include all stages. Can you add these?"

AUTHOR RESPONSE: We thank the reviewer for this suggestion. We have reviewed Figure 1 and have added the headings 'Stage 2a' and 'Stage 2b'. All the stages are now included.

REVIEWER COMMENT 3.4: "Stage 1b - Design: Informed consent? Signed consent forms are mentioned later but what about information sheets? Who will conduct the semi-structured interviews? How long will the interviews be?"

AUTHOR RESPONSE: In light of the reviewer's suggestion, we have now added the following on page 11:

"To promote consistency, interviews will be conducted by one experienced qualitative researcher [AMK]. AMK (PhD), is a university-based qualitative researcher with an interest in health services research but no specialist knowledge of, or involvement in, the ambulance or emergency medical services. Before agreeing to take part, participants will be given a participant information sheet detailing the research. Written informed consent will be obtained [by AMK] before the interviews. No non-participants will be present during the interviews. Interviews will take place at a time and place convenient to the participant. It is anticipated they ; and will last 60-90 minutes."
."

REVIEWER COMMENT 3.5: "Stage 2b – Recruitment: General practices are recruited from one region, will this mean that there is an issue with generalisability?"

AUTHOR RESPONSE: A reasonably large number (up to 58) of general practices from across the NorthWest regions of England will act as patient identification centres for the DCE and we aim to recruit a representative sample. We do agree with the reviewer though that because recruitment is restricted to one region this might impact on the generalisability of the project's findings. To reflect this point, we now explicitly note this within the limitation section to the manuscript (please see page 2). We thank the reviewer for their comment.

REVIEWER COMMENT 3.6: "Analysis: Can you provide a bit more information on outcome and explanatory variables for the random effects logistic regression models."

AUTHOR RESPONSE: We have now revised the manuscript to provide this additional detail on pages 15 and 16 of our manuscript:

"Analysis: Data for the different seizure vignettes will be analysed separately using random effects logit regression models that allows for multiple observations from the same respondent. This will determine the importance of the different care attributes for preference and direction of effect (as indicated by the significance of the coefficients and their size). Comparison between vignettes will depend on attribute and level specification of each experiment. Analyses will be completed with and without respondents failing a 'rationality' test.

Marginal rates of substitution (the rate at which respondents are willing to give up a unit change in one attribute in exchange for a unit change in another while maintaining the same level of utility) will be calculated using each quantitative attribute as the value attribute with bootstrapped confidence intervals (1000 reps). Analyses will be completed with and without respondents failing a 'rationality' test.

Heterogeneity in preference will be assessed, using subgroup analysis and log likelihood ratio testing, and the model specified accordingly.⁽⁹¹⁾ A-priori subgroups of interest are: (1) emergency care for epilepsy in the past 12-months (paramedic and/or ED visit), (2)

The influence of participants' characteristics on preference, including whether they had visited an ED in the prior 12 months and whether they were a patient or a carer, will be examined as a secondary analysis to allow us to determine what differences exist. We shall also conduct subgroup analyses to explore potential differences in preference based on: (1) social deprivation (with participants grouped according to how socially deprived the area is within which they live); (3) whether or not they have seen an epilepsy specialist within the prior 12 months (which they should have according to NICE guidelines [89] given they will all have uncontrolled epilepsy); and (4) whether they or the person they know in the case of carers, self-reports an intellectual disability. The influence of participants' characteristics on preference, including whether they had visited an ED in the prior 12 months, whether they were a patient or carer, social deprivation, and presence of an intellectual impairment, will be examined as a secondary analysis.

Marginal rates of substitution (the rate at which respondents are willing to give up a unit change in one attribute in exchange for a unit change in another while maintaining the same level of utility) will be calculated using each quantitative attribute as the value attribute with bootstrapped confidence intervals (1000 reps).. Comparison between vignettes will depend on attribute and level specification of each experiment.

'Utility scores' for different ACP configurations and significantly different populations will be calculated and ranked accordingly. Their uptake under different seizure scenarios will be estimated.

All analyses will be conducted in STATA 11 (StataCorp, College Station, TX, USA)."

REVIEWER COMMENT 3.7: "Table 1 - Not clear why patients need to be 18+ but carers 16+; Are views of children with epilepsy (or their carers) represented?"

AUTHOR RESPONSE: This project focuses on adults with epilepsy. In light of the reviewer's comment, we now explicitly note the focus of the project at the start of our manuscript and within its title. As per NICE guidelines for epilepsy, adults are defined as those aged 18 years and older. In line with this, the (Quality and Outcome Framework, QOF) registers held within primary care of PWE, by which we shall identify potential participants for the DCE, are restricted to people aged 18+.

We did not consider it possible, via this project, to also account for the views of children or their carers. Not only are the epilepsy care models for adults and children/ younger people in UK quite different, but the momentum regarding the use of ACPs following seizures has focused primarily on adults. This is partly because evidence on the deficiencies in emergency care relates to adults not children and because the non-conveyance policies of ambulance services for children following seizures tend to be more restrictive.

Whilst PWE need to be aged 18+ to participate in the DCE, informal carers can participate as long as they are 16+. The reason the age criteria is slightly lower for this category of participants is because preparatory PPI activities for the project and clinical experience indicates that in some instances, young carers are often present, key in the support of adults with epilepsy, but that they would be automatically excluded if the eligibility criteria stipulated the carers also needed to be 18+. The lower age criteria therefore goes some way to try to capture the views of this group on ACPs.

We are aware that informal carers may sometimes be even younger than 16. In England there is currently no statute governing a child's right to consent to take part in research (other than a Clinical Trial of an Investigational Medicinal Product where it is 16). In the absence of law relating specifically to research, it is commonly assumed that the principle of 'Gillick competence' can be applied not only to consent for treatment, but also to consent for research. A child/young person's right to give consent is though dependent upon their capacity to understand the specific circumstances and the details of the research being proposed. This highlights the importance of the invitation letters and participant information sheets that are to be sent to potential participants by the GPs in our project and the need for them to be accessible.

Such documents require the right style to engage the reader and help to establish a rapport by fully respecting their perspective and exploring all of the issues that are important to them. To minimise the burden on GP participant identification sites, we considered it necessary for only one set of recruitment documents to be used, rather than expecting them to tailor the information distributed depending on participant age. Whilst we considered it feasible to generate a single set of recruitment documents that would be accessible and appropriate to those aged 16+, we did not consider it preferable to generate a single set documents that could be used with those even younger. One risk that might occur if this was done is that in making them accessible for young children, the language and presentation could be perceived by older patients as trivialising the subject matter and so negatively impacting upon their recruitment.

REVIEWER COMMENT 3.8: "Group 2 carers the first bullet point lives with the patient group. "

AUTHOR RESPONSE: In Table 1, we have moved the first bullet point 'Lives in the North West of England' for 'Group 2 carers inclusion criteria' to the patient group (please see page 21).

REVIEWER COMMENT 3.9: "Overall there are a lot of acronyms. A list of acronyms may help. Can Epilepsy Action be in full? Initials for research team members could be in brackets to differentiate from acronyms."

AUTHOR RESPONSE: We are grateful to the reviewer for this suggestion and so we have added a list of abbreviations after the Abstract on page 3. To differentiate abbreviations, the initials of the research team members are now in square brackets, such as "The panel will include expertise in

DCEs [EH], emergency medicine [SG], neurology [AM], paramedical science [MJ], general practice [JD], and psychology [AN], as well as having service user representation.”

Epilepsy Action is now written in full throughout the manuscript.

REVIEWER COMMENT 3.10: “Citations should be after punctuation but appears in text both before and after.”

AUTHOR RESPONSE: Thank you for highlighting this. All citations now appear after punctuation in the text.

VERSION 2 – REVIEW

REVIEWER	Wanqing Zhang UNC Chapel Hill USA
REVIEW RETURNED	15-Aug-2019

GENERAL COMMENTS	Thank you for the opportunity to review and comment on the revisions to this article. I appreciate the care that the authors took in addressing the extensive comments made to the original version. The only suggestion is to use mixed effects model incorporation of random effects (both intercept and slope), instead of using random effects logit regression, as the attributes described below should be specified as fixed effects: 1) emergency care for epilepsy in the past 12-months (paramedic and/or ED visit), (2) social deprivation (with participants grouped according to how socially deprived the area is within which they live); (3) whether or not they have seen an epilepsy specialist within the prior 12 months (which they should have according to NICE guidelines given they will all have uncontrolled epilepsy); and (4) whether they or the person they know in the case of carers, self-reports an intellectual disability.
--

REVIEWER	Elizabeth Cecil Imperial College London, UK
REVIEW RETURNED	13-Aug-2019

GENERAL COMMENTS	All my previous comments) have been addressed One query - should Aims c) on page 8, relate to Stage 1c and not 1b
--

VERSION 2 – AUTHOR RESPONSE

REVIEWER 2

REVIEWER COMMENT 2.1: “...I appreciate the care that the authors took in addressing the extensive comments made to the original version. The only suggestion is to use mixed effects model incorporation of random effects (both intercept and slope), instead of using random effects logit regression, as the attributes described below should be specified as fixed effects:

1) emergency care for epilepsy in the past 12-months (paramedic and/or ED visit), (2) social deprivation (with participants grouped according to how socially deprived the area is within which they live); (3) whether or not they have seen an epilepsy specialist within the prior 12 months (which they should have according to NICE guidelines given they will all have uncontrolled epilepsy); and (4) whether they or the person they know in the case of carers, self-reports an intellectual disability..”

AUTHOR RESPONSE: Many thanks to the reviewer for their suggestion regarding the DCE model specification. We have reviewed our protocol, in consultation with one of our scientific advisors, and amended the manuscript to detail our revised plan.

We do not plan to include the respondents’ characteristics directly into the model because we are modelling

respondents’ stated choice between two or more alternatives in a choice task, and within each choice task

respondents’ characteristics are constant.

We have, however, revised our approach and plan to use an error components logit model, which is stronger than random-effects logit because it will model the panel of responses from respondents rather

than just individual choice task. We will further explore preference heterogeneity by estimating a mixed logit

model (with random parameters for the attributes); and, as we are interested specifically in how preference

heterogeneity is linked to respondents’ characteristics we will do this through interaction terms between the

attributes and respondents’ characteristics. We therefore now note the following within our revised manuscript:

“Analysis: Data for the different seizure vignettes will be analysed separately using an error components logit model that allows for a panel of responses from the same respondent. This will determine the importance of the different care attributes for preference and direction of effect (as indicated by the significance of the coefficients and their size). Comparison between vignettes will depend on attribute and level specification of each experiment. Analyses will be completed with and without respondents failing a ‘rationality’ test.

Marginal rates of substitution (the rate at which respondents are willing to give up a unit change in one attribute in exchange for a unit change in another while maintaining the same level of utility) will

be calculated using each quantitative attribute as the value attribute with bootstrapped confidence intervals (1000 reps).

Preference heterogeneity will be assessed by estimating a mixed logit model, which will include random parameters for the attributes, and interactions between attributes and respondent characteristics.(91) A-priori characteristics of interest are: (1) emergency care for epilepsy in the past 12-months (paramedic and/or ED visit), (2) social deprivation (with participants grouped according to how socially deprived the area is within which they live); (3) whether or not they have seen an epilepsy specialist within the prior 12 months (which they should have according to NICE guidelines [89] given they will all have uncontrolled epilepsy); and (4) whether they or the person they know in the case of carers, self-reports an intellectual disability.

'Utility scores' for different ACP configurations and significantly different populations will be calculated and ranked accordingly. Their uptake under different seizure scenarios will be estimated.

All analyses will be conducted in STATA 11 (StataCorp, College Station, TX, USA)."

REVIEWER 3

REVIWER COMMENT 3.1: "One query - should Aims c) on page 8, relate to Stage 1c and not 1b."

AUTHOR RESPONSE: We can confirm that Aim c – namely, to identify the attributes of post-seizure emergency care that PWE and their carers consider important and identify if this differs by the context in

which the seizure occurs – will as currently stated seek to be addressed by Stage 1b (qualitative interviews

and associated analysis. We have thus not made any changes to the manuscript in light of this comment.

VERSION 3 – REVIEW

REVIEWER	Wanqing Zhang UNC Chapel Hill USA
REVIEW RETURNED	26-Sep-2019
GENERAL COMMENTS	Overall this is a very interesting article that should be published in the journal